# PRIMITIVE EMBEDDINGS FOR GENERATIVE MODELING IN INVERSE LITHOGRAPHY

## ABSTRACT

During the manufacturing process of integrated circuits, we require a mask in order to print a certain wafer design. Predicting this mask design is a complex task in the field of inverse lithography. The mapping from wafer to the mask design is ill-posed and requires solving a non-convex optimization task, having multiple potential solutions. Any difference in the setup of the problem (e.g. initialization, patching, or a different discretization scheme) tends to generate inconsistencies. The designed wafer features generally consist of a defined set of basic objects (primitives). Larger features can be built by transforming and aggregating these primitives. Following these observations, we propose a holistic generative approach that utilizes primitive embeddings. We use a model that encodes primitives per type, embeds positional information and then aggregates the feature information. A variational inference approach is then used to take samples of these encodings in the latent space. The samples are transformed by normalizing flows that try to recover the postulated distribution before constructing a mask design. A generative model predicts one of the best mask designs, avoiding inconsistencies and thus allowing for a flexible design approach. Finally we introduce a novel scoring method to fit this probabilistic setup. We assess the performance of this approach for a simplified inverse lithography setup. The main purpose of this study is to investigate the use of primitive modeling in inverse lithography tasks. Although we are not yet able to reach benchmark accuracy within this new setup, results are promising from an application point of view.

## 1 INTRODUCTION

Integrated circuits lie at the basis of all digital technology. The microchips are typically manufactured on a silicon wafer, with small features ($\mathcal{O}(10\text{nm})$) imprinted to make up components like transistors. The lithography process is a key step in the manufacturing of these integrated circuits (IC). It involves the transfer of a chip design (encoded by a reticle mask) onto a light sensitive material (photoresist) through a series of lenses or mirrors. Due to optical limitations, the reticle mask needs to be designed knowing the characteristics of light, lens, mask and photoresist. Sub-resolution assist features (SRAFs) need to be added to ensure that the masks are robust with regards to (stochastic) variations of the light source, optical and resist properties (Shen et al. (2011),Shen (2018)). The desired wafer design is also known, leaving the mask design to be optimized in order to produce the desired wafer features. This process is known as inverse lithography technology (ILT) and has for a long time been one of the biggest challenges in semiconductor manufacturing.

Optimizing the masks for a full chip design is highly computationally demanding. Typically, the mask optimization is done iteratively by minimizing a pattern fidelity loss, subject to manufacturability constraints. To provide a faster solution, deep learning surrogate models are trained in a supervised manner and used as fast surrogate models for subtasks of predicting a continuous transmission masks CTM (see e.g. Yang et al. (2018)). However, the ILT optimization problem is ill-posed, with multiple solution masks producing the same target on the wafer. A naive deep learning model will therefore lead to poor mask predictions, averaging over all possible outcomes (Lugmayr et al. (2020),Pathak et al. (2016)). Generative models might help by capturing a joint probability distribution $p(\phi, z)$ on observations $\phi$ and their latent parameterization $z$ (Kingma et al., 2019).

Existing methods for mask design typically treat the entire chip as one entity. Patching (dividing the chip into smaller parts) is used to make the optimization problem tractable, by providing smaller data sizes. However, using such patches comes at the cost of inconsistency at borders between neighboring patches, due to the ill-posed problem. In addition, changing a single feature still requires remapping the entire patch. A more flexible, modular setup of chip design would be beneficial. We propose to integrate a so-called primitive modeling setup with a variational auto-encoding (VAE) architecture. We adopt a separable feature encoding scheme, to allow for flexible target generation and easy domain validation. Next we use generative modeling to deal with the one-to-many mapping of target to masks, while introducing a novel scoring method to fit this probabilistic setup.

## 2 PROBLEM STATEMENT

Inverse lithography is the process of generating a mask to produce a desired target. Let $\phi \in \mathcal{S}_\phi$ be the desired target image, $\omega \in \mathcal{S}_\omega$ be the wafer exposure image, and $\psi \in \mathcal{S}_\psi$ be a mask pattern (Pang, 2021). The dimensions of the target and mask image are $D_{\text{target}}$ and $D_{\text{mask}}$ respectively. Also, let $F : \mathcal{S}_\psi \to \mathcal{S}_\omega$ be the forward model that maps a mask image to the target image. $F$ contains an optical model $H$, expressing the low-pass filtering effects of the lens as light passes through it, and a resist model $T$ (Shen et al., 2011), which is essentially a thresholding function caused by the local activation of resist chemistry from the energy deposited on the wafer. Using these definitions, we have:

$$\omega = F(\psi) = T\left(|H * \psi|^2\right) + e\,, \tag{1}$$

where $e$ captures stochastic disturbances such as the number of photons that hit the resist material Pang (2021). The goal is to find a mask that brings the exposed wafer target $\omega$ as close as possible to the desired target $\phi$. Due to the optical resolution limitations, assist features have to be added to mask designs that sharpen the projection of the main features on the photoresist material. The contribution of these assist features is kept below the activation energy level of the resist, meaning they do not contribute to direct features on the wafer target (Shen et al., 2011). The placement of these assist features is non-trivial: the forward model is a one-to-many mapping (Pang, 2021). In other words, $F$ is not a bijective function so the inverse of $F$ does not exist. The state-of-the-art approach to finding an appropriate mask is to define a loss function, typically a mean squared error, characterizing the dissimilarity between the exposed wafer and the desired target. This loss is then optimized with respect to a continuous relaxation of mask space, known as continuous transmission masks (CTM), with nonlinear least-squares solvers.

We characterize the problem as maximum a posteriori estimation in a probabilistic generative model:

$$\psi^* = \underset{\psi \in \mathcal{S}_\psi}{\arg\max}\ p(\phi \mid \omega)\, p(\omega \mid \psi)\, p(\psi)\,, \tag{2}$$

where $p(\omega \mid \psi)$ describes the likelihood of wafer exposures given masks and $p(\psi)$ is a prior distribution over masks. The distribution $p(\phi \mid \omega)$ is essentially another likelihood function that is high when the exposed wafer target is similar to the desired target. In the next sections, we explain how we model these distributions and how we derive an effective inference procedure.

## 3 PRIMITIVE MODELING

The space of possible masks $\mathcal{S}_\phi$ for full target images is vast and searching it naively with an optimization procedure will take a long time. We propose a smaller representation of the target space that we hypothesize will make the inference problem easier.

Note that wafer targets $\phi$ are combinations of simple (i.e., non-self-intersecting) axis-aligned polygons (Cecil et al., 2022). We refer to an individual polygon as a *feature* $\mathcal{G}$.

**Theorem 3.1.** *Any simple axis-aligned polygon $\mathcal{G}$ can be expressed as a finite union of rectangles.*

The proof is a classical result in computational geometry (De Berg et al., 2008). Since targets are simple axis-aligned rectangles, targets can be expressed as finite unions of rectangles. It follows directly that the space of targets is covered by the space of finite unions of rectangles. But the two

spaces are not equivalent. The lithography system is diffraction-limited, which means two polygons in the target must have a minimal distance between them. If the two polygons are too close to each other, then the resulting mask is indistinguishable from the mask of a single polygon. In other words, if the gap is too small, it is as if there would be no gap. Under finite unions of rectangles it is possible to construct simple axis-aligned polygons that violate that constraint. These would be elements from the space of finite unions of rectangles that are not part of $\mathcal{S}_\phi$. This fact prevents us from generating targets by optimizing over arbitrary finite unions of rectangles.

We will instead represent targets as combinations of primitives.

**Definition 1.** *A primitive $\phi_p$ is a feature (i.e., a single simple axis-aligned polygon) from actual wafer targets used in industry, represented as a binary image. In this image, white pixels represent the interior of the polygon and black the exterior. As such, pixels represent the rectangles that partition the polygon.*

Figure 1 shows three examples of these primitives on the left. We construct targets by combining smaller primitives on a larger blank image through spatial convolution:

$$\sum_{i=0}^{N} \phi_p * \delta_{x_i, y_i} \mapsto \phi \,, \tag{3}$$

where $\delta_{x_i, y_i}$ is a two-dimensional unit impulse located at the point $(x_i, y_i)$. From modeling of the physics (section 1) we know that the optical filtering kernel has limited field of view. This observation motivates the primitive modeling approach by suggesting that features are composed of nearby primitives.

Later on, we shall use variational autoencoders that encode and decode primitives. Training a model that encodes these primitives independently, and subsequently aggregates and decodes the embeddings has several advantages. First, the model is suitable for any desired target that lies within the pre-defined primitive domain, allowing easy validation of target design. Secondly, the entire mask need not be recomputed when changing a single feature or primitive. Instead the modular setup allows to recompute just the local area around the change, with less risk of introducing inconsistencies. Finally, primitive aggregation can offer full field of view to the decoder, by aggregating all encoded primitives in the lower-dimensional encoding manifold. When reconstructing a mask or wafer target, the decoder can thus select the relevant encoded primitives based on their associated positions. This reduces inconsistencies from patching.

## 4 GENERATIVE MODELING

The goal of generative modeling is to find a model that best explains the occurrence of observations ($\phi$), i.e. maximizes the evidence $p(\phi)$. Typically we find it more convenient to optimize for the log-evidence instead, which leaves us to finding a tractable expression for $\ln p(\phi)$.

To this end, we introduce a latent variable $z$ to the generative model and alters its factorization to be:

$$p(\psi, \omega, \phi, z) = p_\sigma(\phi \mid \omega) p_\theta(\psi, \omega \mid z) p(z) \,. \tag{4}$$

We introduce a variational model over masks and the latent variable $z$ given the target $\phi$:

$$q(\psi, z \mid \phi) = q(\psi \mid z) q(z \mid \phi) \,. \tag{5}$$

Then we formulate an ELBO by lower bounding the log-evidence term (Kingma et al., 2016):

$$\ln p(\phi) \geq \mathbb{E}_{q(\psi, z \mid \phi)} \left[ \ln \int p(\psi, \omega, \phi, z) \mathrm{d}\omega - \ln q(\psi, z \mid \phi) \right] \tag{6}$$

$$= \mathbb{E}_{q(\psi, z \mid \phi)} \left[ \ln \int p(\psi, \omega, \phi | z) \mathrm{d}\omega \right] - D_{\mathrm{KL}}[q(z|\phi)||p(z)] - \mathbb{E}_{q(\psi \mid z)} \left[ \ln q(\psi \mid z) \right] \tag{7}$$

$$= \mathbb{E}_{q(\psi, z|\phi)} \left[ \ln \int p_\sigma(\phi|\omega) p_\theta(\psi, \omega|z) \mathrm{d}\omega \right] - D_{\mathrm{KL}}[q(z|\phi)||p(z)] - \mathbb{E}_{q(\psi|z)} \left[ \ln q(\psi|z) \right]. \tag{8}$$

To tackle the integral over $\omega$, we introduce an importance sampling step over a distribution $r(\omega \mid z)$:

$$\int p_\sigma(\phi \mid \omega) p_\theta(\psi, \omega \mid z) \mathrm{d}\omega = \mathbb{E}_{\omega \sim r(\omega \mid z)} \left[ \frac{p_\sigma(\phi \mid \omega) p_\theta(\psi, \omega \mid z)}{r(\omega \mid z)} \right] \,. \tag{9}$$

This expectation is plugged into Eq. 8, which - through Jensen's inequality - yields:

$$\ln p(\phi) \geq \mathbb{E}_{q(\psi, z \mid \phi)} \mathbb{E}_{\omega \sim r(\omega \mid z)} \Big[ \ln \frac{p_\sigma(\phi \mid \omega) p_\theta(\psi, \omega \mid z)}{r(\omega \mid z)} \Big]$$
$$- D_{\mathrm{KL}}[q(z \mid \phi) || p(z)] - \mathbb{E}_{q(\psi \mid z)} \big[ \ln q(\psi \mid z) \big] \tag{10}$$

Given deterministic (neural networks) $\omega = g_\omega(z)$ and $\psi = g_\psi(z)$, this reduces to:

$$\ln p(\phi) \geq \underbrace{\mathbb{E}_{z \sim q(z \mid \phi)} \big[ \ln p_\theta(\psi = g_\psi(z), \omega = g_\omega(z)) \big]}_{\text{joint plausibility generated mask and wafer projection}} + \underbrace{\mathbb{E}_{z \sim q(z \mid \phi)} \big[ \ln p_\sigma(\phi | z) \big]}_{\text{data fidelity / accuracy}}$$
$$- D_{\mathrm{KL}} \big[ q(z|\phi) || p(z) \big] - \mathbb{E}_{q(\psi \mid z)} \big[ \ln q(\psi \mid z) \big]. \tag{11}$$

Note that the entropy over $q(\psi \mid z)$ is 0 because $\psi$ is generated deterministically from $z$. For the accuracy term, we take the 2-norm between the generated wafer projection $\omega$ and the desired wafer projection $\phi$:

$$\ln p(\phi) \geq \mathbb{E}_{z \sim q(z|\phi)} \big[ \ln p_\theta \big( g_\psi(z), g_\omega(z) \big) \big] - \mathbb{E}_{z \sim q(z|\phi)} \Big[ \frac{1}{2\sigma^2} || \phi - g_\omega(z) ||_2^2 \Big] - D_{\mathrm{KL}} \big[ q(z|\phi) || p(z) \big]. \tag{12}$$

As stated before in Section 2, the mapping $\phi \rightarrow \psi$ is one-to-many and ill-defined (we may not know all valid masks for a given $\phi$). For these reasons, we cannot use a naive pixelwise loss function for mask reconstruction, neither can we construct a categorical-like loss. Instead, we derived a loss term which is reminiscent to the adversarial loss in GANs (Goodfellow et al., 2014) and the loss value in membership inference (Hu & Pang, 2023). This loss term (first component of the bound in equation 12) is the joint plausibility of the generated mask and wafer projection. To evaluate it, we propose training a standalone model that learns a proxy to the joint distribution $p(\omega, \psi)$. This learned distribution can then be used to evaluate predicted samples $p(g_\psi(z), g_\omega(z))$, ensuring consistency between predicted mask and wafer target (effectively the mask prediction model should learn a joint latent distribution). Having such a probabilistic loss score nicely fits the context of generative modeling and matches the difficulties that come with a one-to-many mapping.

The second component of the bound in equation 12 is a data fidelity term that enforces the output by the generator to have high likelihood, acting as a reconstruction error. The third component can be thought of as a regularization term on the latent posterior, forcing it to approximate the postulated prior distribution.

Different setups can be used for the generative and inference model; we use the variational auto-encoder (Kingma et al. (2019), Kingma & Welling (2013)), using a Gaussian prior on $p(z)$.

## 4.1 Transformation of latent distribution

The better the posterior distribution $p(z \mid \phi)$ approximates the true distribution, the better we will be able to optimize the model parameters (Van Den Berg et al., 2018). Unfortunately, a Gaussian prior has limited descriptive power and it may not fully capture a complex (e.g. multi-modal) distribution, which hampers performance. The authors from (Rezende & Mohamed, 2015) tackle this problem by using normalizing flows: a series of invertible transformations (using learned parameters) acting on the prior. This effectively increases the expressiveness of the prior by constructing complex distributions out of a Gaussian, such that the true posterior can be recovered. A special case of this is Sylvester flow (Van Den Berg et al., 2018), which uses a set of mathematical tricks to ensure easy optimization of the (flow) model. After $K$ flow operations, we can rewrite the new, more complex latent distribution as:

$$q_K(z_K) = q_0(z_0) \prod_{k=1}^{K} \left| \frac{\partial f^{-1}(z_{k-1})}{\partial z_{k-1}} \right|. \tag{13}$$

Mathematical details can be found in Appendix A.2. This new distribution $q_K(z_K)$ allows for a more expressive posterior approximation. Replacing $q$ with $q_K$, we can rewrite the last term of ELBO from equation 12 as follows:

$$\mathbb{E}_{z_K \sim q_K} [q_K(z_K \mid \phi) - \ln p(z)] = \mathbb{E}_{z_0 \sim q_0} [\ln q_0(z_0 \mid \phi) \cdot \mathcal{P} - \ln p(z)] \tag{14}$$
$$= D_{KL}(q_0(z_0 \mid \phi) || p(z)) + \mathbb{E}_{z_0 \sim q_0} [\ln \mathcal{P}] \tag{15}$$

Figure 1: High level model overview. On the left, three stacks of images are shown for three different types of primitives. Each stack is passed through the primitive embedding part (grey box) separately, but for illustrative purposes this is only shown once. The resulting aggregated vectors per primitive type are concatenated and passed on to the rest of the model.

where we define $\mathcal{P}$ as:

$$\mathcal{P} = \prod_{k=1}^{K} \left| \frac{\partial f^{-1}(z_{k-1})}{\partial z_{k-1}} \right|. \tag{16}$$

The first term in equation 15 is the KL-divergence between the simple base distribution $q_0$ and the Gaussian prior $p(z)$. The second term of equation 15 accounts for the change of volume that comes with the flow transformations. Having rewritten ELBO by incorporating normalizing flows, we drive the sampling distribution towards a Gaussian. Meanwhile, the approximate posterior for the latent variables is pushed to be as close to the real posterior as possible through optimizing the flow variables.

## 4.2 LOSS FUNCTION

Putting everything together, we arrive at the following expression to optimize our bound, by minimizing the loss function $\mathcal{L}$:

$$-\mathcal{L} = \mathbb{E}_{z \sim q(z \mid \phi)} \left[ \ln p_\theta \big( g_\psi(z), g_\omega(z) \big) \right] - \mathbb{E}_{z \sim q(z \mid \phi)} \left[ \frac{1}{2\sigma^2} ||\phi - g_\omega(z)||_2^2 \right]$$
$$- D_{KL} \left[ q_0(z_0 \mid \phi) || p(z) \right] + \mathbb{E}_{z_0 \sim q_0} \left[ \ln \mathcal{P} \right]. \tag{17}$$

With the first term a joint plausibility score based on a learned distribution $p(\omega, \psi)$ to ensure mask and wafer target consistency, and the second term a pixel-wise loss accounting for the data fidelity term. $p(z)$ is a Gaussian prior and $\mathcal{P}$ accounts for the flow transformations. The individual terms will be weighted to account for any imbalances.

## 5 MODEL ARCHITECTURE

Our main contribution is the mask prediction model based on primitive embeddings. This model encodes primitives and their positions, aggregates the embeddings and based on this generates an output. A high-level overview of our proposed model architecture is given in Fig. 1.

### 5.1 PRIMITIVE EMBEDDING

One data sample $\phi$ consists of a set of primitives that together form a set of one or more features in the field of view. Primitives of the separate types are grouped together. Each primitive has a specific position with coordinates $(x, y)$ within the field of view, as in (Dosovitskiy et al., 2020). These coordinates are normalized and embedded into a vector using a sinusoidal encoding scheme, similar to (Vaswani et al., 2017). The authors from (Annamoradnejad & Zoghi, 2024) show that learning of textual structural information is improved when an overview of the full text is given, in addition to individual sentences. To simulate this, we give the model positional awareness by taking the relative position of each primitive with regard to all other primitives in the field of view. These distances are

normalized, encoded and then summed into two final relative embedding vectors (for $x$ and $y$). This gives four position vectors per primitive. Besides, we have an image that uniquely represents its shape, which is embedded via the inference model $q(z \mid \phi)$ via a series of convolutions. At different steps throughout the embedding, the position vectors are aggregated with the intermediate image embeddings. This ensures that positional information is well represented and integrated in the final primitive embedding.

The separate primitive types are fed through the encoding module separately (see fig. 1, however encoder weights are shared. The embedded primitive vectors are summed per type, leaving one vector for each type of primitives. These are aggregated through a simple concatenation and then fed through a fully connected block, leaving a single vector $h$. From this a mean $\mu$ and standard deviation $\sigma$ vector are extracted through a linear transformation of $h$. These vectors $\mu_0$ and $\sigma_0$ describe the posterior distribution $q_0(z \mid \phi)$.

## 5.2 Sampling and flow

From $q_0(z_0 \mid \phi)$, we obtain a set of latent variables $z_{0,i} \forall i \in \{0, \ldots, 63\}$. Random sample variables are drawn from a normal distribution $\epsilon \sim \mathcal{N}(\epsilon \mid 0, 1)$. We use the reparameterization trick to compute $z_0 = \mu_0 + \sigma_0 \cdot \epsilon$ (Kingma & Welling, 2013). The sampled latent variables $z_0 \sim q_0(z_0 \mid \phi)$ are transformed by the flow model into $z_K = f(z_0)$. Latent variables $z_K$ are then passed on to the generative model $p(\phi \mid z_K)$.

## 5.3 Decoding and translation

The decoder consists of two transposed convolutional blocks that have the exact same architecture and are trained simultaneously. The first decoder produces a wafer target $\omega = g_\omega(z)$ with all primitives placed at their corresponding position in the field of view. Reconstructing the original image from the primitives, forces the model to learn a meaningful latent posterior distribution $p(z \mid \phi)$. Besides, we can validate model output during inference, by evaluating whether the (known) mapping from primitives to image is performed well. The second decoder predicts the desired mask image $\psi = g_\psi(z)$. While there is only one valid wafer target $\omega$ given $\phi_k$, there may be a set of $N$ mask images $\psi^\phi = \{\psi_1^\phi, \psi_2^\phi, \ldots, \psi_N^\phi\}$ associated with this wafer target $\omega$. Each of these mask images is an equally valid output from the second decoder.

## 5.4 Error prediction model

A high-level implementation of the classifier/regression model that is used for predicting the loss score is given in Fig. 2a. As input we use pairs of a wafer target image $\omega$ with an associated mask image $\psi$, which are embedded separately. To guide the model, we use a third input channel, for which we take the absolute difference $|\omega - \psi|$. This gives an indication of the correspondence between the two images, since the main features of the wafer target are also present in the mask. Each input channel has its separate encoder, and the embedded vectors are concatenated and further embedded into one latent vector. The model has two output heads to give both a classification and regression score. They have the same number of linear layers for decoding, however the classifier head uses Sigmoid activation, while the regressor head uses ReLU activation. They are trained using binary cross-entropy (BCE) and mean squared error (MSE) loss respectively.

## 6 Experimental setup

A synthetic data set is used to train, evaluate and test the model. According to the primitive modeling setup described in Section 3, a hierarchical data generation method is used to generate target images. Details can be found in the supplementary material. The resulting target images are passed through an iterative solver to compute the continuous transmission mask images. Settings have been determined to create a simple but realistic setup, the details of which can be found in Appendix A.4. A dataset of 3000 images is synthesized and split 80-20 into a training and validation set.

Model optimization is done using Adam optimization method (Kingma & Ba, 2014). The error prediction model and mask prediction model are trained for a maximum of 500 and 1000 epochs respectively. Early stopping is used when convergence was reached and/or training became unstable.

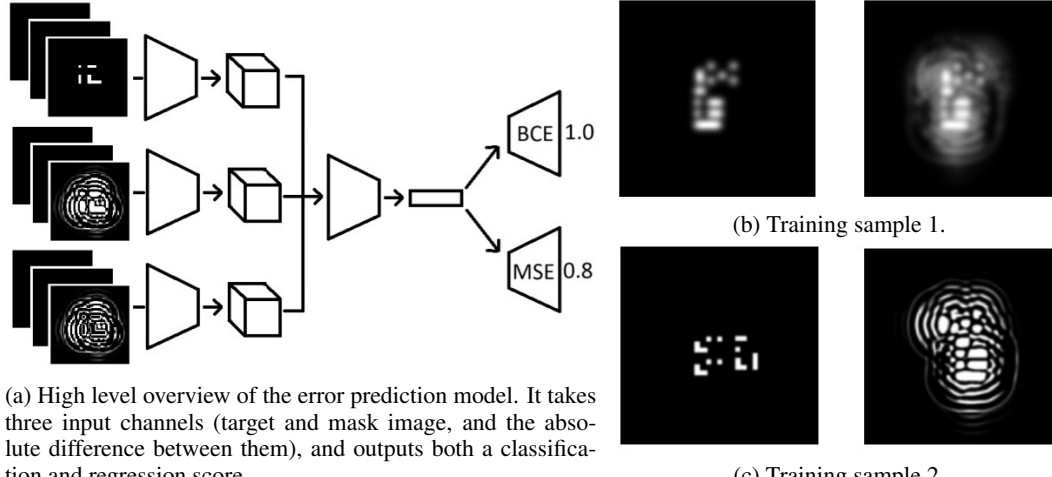

(a) High level overview of the error prediction model. It takes three input channels (target and mask image, and the absolute difference between them), and outputs both a classification and regression score.

(b) Training sample 1.

(c) Training sample 2.

Figure 2: On the left (figure 2a) an overview of the error prediction model architecture is given. On the right, two training samples for this model are given. 2b shows a correct but blurry match of target and mask while 2c shows a mismatched sample.

To evaluate model performance, we consider the following general objectives: 1) the predicted mask must be valid, 2) all valid masks should have equal likelihood and 3) the final predictions should be highly accurate. Experiments to test the objectives are reported in Section 7.

The error prediction model is trained first, to be used as scoring model later. For each pass through the model, wafer target images are rematched to a random other mask image in the dataset with 40% probability. Additionally masks and wafer targets are blurred with a 50% probability, using a Gaussian filter operation with varying standard deviation. This simulates the blurry predictions that are made during the first stages of training of the mask prediction model. Omitting this step would lead to the model only predicting meaningful error scores for fully trained model predictions (which has no use during early stages of training). The classifier output is a simple indicator of whether a wafer target is associated with its original mask (1), or has been rematched (0). Since this scoring is binary, the classifier head is trained with binary cross-entropy loss. The ground truth for regression score is a normalized MSE-loss between the wafer target and the result of passing the associated mask through the optics and resist model. Depending on the possible wafer target-mask rematching and applied blurring, this score varies between 0 and 1, where 0 indicates a perfect match. The regressor head is trained using mean squared error loss. Optimization settings for training of the error prediction model can be found in Appendix A.7.

After training the error prediction model, it can be used to train the mask prediction model. Only regression scores are used in model optimization, with exponentially increasing weights while decaying the weight of MSE loss to ensure steady convergence. Note that while the regression scores are required for mask prediction, they also steer wafer target prediction towards probable outcomes. On a high level, we can view the training approach as two-folded: first we force wafer target prediction towards ground truth. Based on the predicted wafer targets, we then push mask predictions to match these. Optimization settings for training of the mask prediction model can be found in Appendix A.8.

# 7 EXPERIMENTS

To test the objectives formulated in Section 6, we conduct experiments to compare the proposed model against a benchmark. Additionally, a report of ablation studies is given in the supplementary material. The test set used consists of samples. We report model sizes as the number of trainable parameters. Note that the error prediction model is applied in inference mode, so using it increases model size but the number of trainable parameters stays the same.

We include image-to-image mapping in some setups, to evaluate the model's ability to handle input

| Model configuration | Prediction error/score | | # params |
| --- | --- | --- | --- |
| | Regression | Classification | |
| A: two-channel input | $0.0399 \pm 0.074$ | 99% | 5.5M |
| B: three-channel input | $0.0239 \pm 0.0418$ | 100% | 8.0M |

Table 1: Results of training regressor and classifier loss model

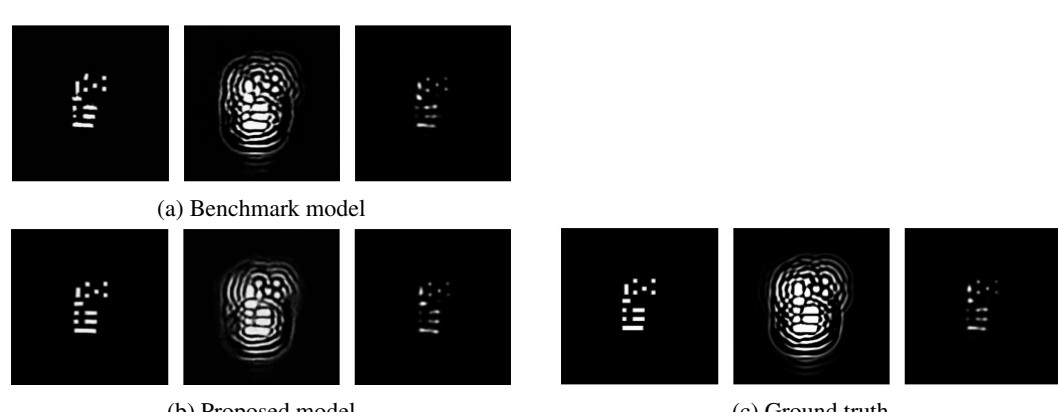

(a) Benchmark model

(b) Proposed model           (c) Ground truth

Figure 3: Results from different models on one data sample. The left column shows the reconstructed wafer target $\omega$ (3a and 3b), or target design $\phi$ (3c). The middle columns shows an associated mask $\psi$ (prediction) and the right columns shows the resulting wafer image $F(\psi)$.

data in the form of primitives and their positions. Please note however, that from an application perspective this image-to-image setup is less desirable, for reasons mentioned in Section 3.

The classification/regression error prediction model must be trained prior to training the mask prediction model. Quantitative results are provided in table 1 for setup A with two inputs (wafer target $\omega$ and mask $\psi$), and setup B with an third input (absolute difference between wafer target and mask). As metric for the regression prediction error, we measure the fraction of absolute difference between predicted and true error, with respect to the true error. The classification prediction score is the percentage of images correctly classified. Scores are averaged over all samples from the test set. Some examples of matched inputs from the test set are provided in Figures 2b and 2c.

Table 2 shows quantitative results of our proposed model (setup A), compared to a benchmark (setup B). The benchmark model deploys a simple encoder-decoder setup to perform the image-to-image mapping. These models are evaluated on reconstruction towards both wafer target $\omega$ and mask $\psi$ domain. The metric used for wafer target reconstruction is the mean absolute error between ground truth $\phi$ and predicted output $\omega$. For the ILT targets (masks), the score metric is the mean absolute error between the wafer target (in low resolution) $\phi_{lr}$ and the result of passing the predicted mask through the optics and resist model $F(\psi)$. This gives an unambiguous way of scoring the predict mask performance. An example of model output from our proposed method is given in Fig. 3, where we show ground truth and predicted outcomes.

## 8 DISCUSSION

An important goal of this study was to perform a mapping from primitive to image domain. While this is beneficial application-wise (wafer design becomes more flexible), this is in fact a complex

| Model configuration | Prediction error | | # params |
| --- | --- | --- | --- |
| | MAE$(\phi, \omega)$ | MAE$(\phi_{lr}, F(\psi))$ | |
| A: proposed | $0.0075 \pm 0.065$ | $0.0026 \pm 0.026$ | 13.9M |
| B: benchmark | $0.0027 \pm 0.033$ | $0.00093 \pm 0.0099$ | 14.3M |

Table 2: Model results compared against benchmark on ILT dataset.

task. Comparing model performance for an image-to-image and primitive setup, we see that the first setup performs best. This suggests a more informative latent distribution i.e. structural information is indeed better captured through images than through primitives and their positions. Still, the primitive model is able to reconstruct masks and wafer targets reasonable well, with the main features present.

For a well-functioning mask reconstruction model, predicted error scores should be accurate. The three-channel error prediction model is good at classifying images, but regression scores could be improved. The setup with two-channel input performs slightly worse. Combining the wafer target and mask image earlier in the model (e.g. during embedding) might help, allowing the model to extract the difference between mask and wafer target.

Appendix A.9 and A.10 report ablation studies to individually test the proposed models' components and an experiment to test our generative setup for the one-to-many mapping respectively.

### 8.1 LIMITATIONS

The convolutional network used is well-suited to capture structural information from images. For our primitive setup however, a different encoder (e.g. graph networks) might be better, in which we provide a categorical parameter (for the shapes) and some coordinates from a uniform distribution.

Aggregation of primitive embedding vectors is currently done via a summing operation. More advanced aggregation techniques can be explored to see what captures best the full latent distribution.

The distribution of input data provided during training of the error prediction model, should correspond to the distribution of of output data of the mask prediction model. We tried to approximate by training on mismatched and blurry data, but this is insufficient. Thus, we provide out-of-distribution data during inference, leading to faulty error predictions. A GAN-like training loop where the mask prediction model generates samples to train the error prediction model might be beneficial. With each cycle, the input distribution of the error prediction model will be more similar to the distribution on which it performs inference.

The dataset used has only a few different features. When using a dataset with randomized features (i.e. no pre-defined set of features), the wafer target reconstruction model was more prone to overfitting during training. We hypothesize that this is because with limited sample size, randomized samples have low structural similarity and the model can easily learn to discern them, i.e. overfit. Using larger datasets will force the model to learn the sample distribution instead of individual samples. Using a dataset with 100 different features, the model did not overfit but wafer target reconstruction was difficult.

## 9 CONCLUSION

In this study we trained a prediction model for mask reconstructions in a lithography application. We proposed using primitive embeddings to allow for more flexibility in designing chip patterns in this setting. This poses a harder challenge as we cannot simply map from image to image domain, which caused a decrease in wafer target reconstruction and mask prediction accuracy. However, we show that mask reconstruction based on primitive embeddings is possible and thereby address some major challenges in inverse lithography.

Additionally, to deal with the one-to-many mapping in the data setup, we applied a generative model. We also trained a separate error prediction model that could deal with the multiple possible outputs in the given context. Although a gain in performance is required before the proposed model can be applied in a real lithography setting, we show that our probabilistic setup works well for variable sample prediction. As such, our proposed model aims to replace the inverse solvers that are typically used to compute mask designs, saving computational time and resources. Finally, we introduced the error prediction model in a very general way and we expect that this idea can be utilized in many different applications.

## REPRODUCIBILITY STATEMENT

We described a way of representing wafer target images $\phi$ as a collection of primitives. Appendix A.3 describes settings to synthesize these wafer targets. The corresponding masks can be obtained using an inverse solver and (simplified) forward model, settings of which are given in appendix A.4. The obtained data set can be used to first train the error prediction model, architecture and optimization settings are provided in appendices A.5 and A.7 respectively. Next, the architecture of the mask prediction model is provided in A.6 and optimization settings for different configurations are given in appendix A.8. After training, the models can be used to construct mask predictions.

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

# A APPENDIX

## A.1 PRIOR DISTRIBUTION IN VARIATIONAL INFERENCE

As described in Section 4, a simple prior for the latent distribution can lead to decreased model performance. This gap in performance stems from the difference between the true and approximate posterior and can be inferred from the log-likelihood function $p(\phi)$ (for the full derivation, see (Tomczak)):

$$\ln p(\phi) = \mathbb{E}_{z \sim q} \left[ \ln p(\phi) \right] \tag{18}$$

$$= \mathbb{E}_{z \sim q} \left[ \ln \frac{p(z \mid \phi) p(\phi)}{p(z \mid \phi)} \right] \tag{19}$$

$$= \mathbb{E}_{z \sim q} \left[ \ln p(\phi \mid z) - \ln \frac{q(z \mid \phi)}{p(z)} + \ln \frac{q(z \mid \phi)}{p(z \mid \phi)} \right] \tag{20}$$

$$= \mathbb{E}_{z \sim q} \left[ \ln p(\phi|z) \right] - KL \left[ q(z \mid \phi) || p(z) \right] + KL \left[ q(z \mid \phi) || p(z \mid \phi) \right]. \tag{21}$$

Now the last component $KL \left[ q(z \mid \phi) || p(z \mid \phi) \right]$ gives a measure of the difference between the true posterior $p(z \mid \phi)$ and the approximate posterior $q(z \mid \phi)$. Compare this with the normal equation for ELBO loss to see that this KL-divergence term is not present there:

$$\ln p(\phi) \geq \mathbb{E}_{z \sim q} \left[ \ln p(\phi \mid z) \right] - KL \left[ q(z \mid \phi) || p(z) \right]. \tag{22}$$

In principle, the $KL \left[ q(z \mid \phi) || p(z \mid \phi) \right]$ is non-negative and thus we can leave it out and obtain the ELBO. However, this term shows that there can be a gap between the ELBO that we optimize for, and the true posterior. As a consequence the set of parameters that maximizes ELBO may be suboptimal and is not necessarily the same as the set of parameters that maximizes $\ln p(\phi)$. Consider having a basic Gaussian prior, and a complex (e.g. multi-modal) true distribution $p(z \mid \phi)$. The approximate posterior distribution will never reach the shape of the true distribution, leading to the gap in estimated model parameters that was derived above. This observation motivates setting a more expressive prior in order to reach a better approximation.

## A.2 FLOW MATHEMATICS

To construct a flow series, we use invertible functions $f_k$ (unrelated to the forward model $F$) of the following form:

$$f_k(z_k) = z_k + Ah(Bz + k + b) \quad \text{where} \quad z_k + 1 \leftarrow f_k(z_k) \tag{23}$$

$A$ and $B$ are matrices that can be learned from data, and are constructed according to the method for Orthogonal Sylvester Normalizing Flow (O-SNF) construction described in (Van Den Berg et al., 2018). These functions are used to construct a series of $K$ flow operations, through which we can transform given set of variables $z_0$:

$$z_K = f(z_0) \quad \text{where} \quad f = f_K \circ \cdots \circ f_2 \circ f_1. \tag{24}$$

The new, more complex distribution formed through the flow can be computed by using a change of variables:

$$q_{\chi,K}(z_K) = q_{\chi,0}(z_0) \prod_{k=1}^{K} \left| \frac{\partial f^{-1}(z_{k-1})}{\partial z_{k-1}} \right|. \tag{25}$$

## A.3 WAFER TARGET DATA GENERATION

Each target image consists of multiple features drawn randomly from a pre-defined set, which are placed randomly across the image, in a non-overlapping fashion. Each feature in turn consists of multiple primitives, mimicking standard patterns across chips. Four different types of primitives have been defined: 1) a unit square with width $w = u$ and height $h = u$; 2) a horizontal rectangle ($w = au, h = u$); 3) a vertical rectangle ($w = u, h = bu$), and 4) a corner-like shape, composed of two touching rectangles (one horizontal and one vertical). The horizontal and vertical rectangle have variable width $w$ and height $h$ respectively, while for the corner shape both $h$ and $w$ can vary.

| Image size | 896 nm |
|---|---|
| Feature unit size | 56 nm |
| Max. feature size | 560 nm |
| Pixel size | 7 nm |

Table 3: Settings for generation of target images

| Image size | [128, 128] px |
|---|---|
| Zero-padding | 96 px |
| Pixel size | 7 nm |
| Optical kernel main lobe | [56, 56] nm |
| Sigmoid rescale resist | 20.0 |
| Threshold resist | 0.30 |
| Nr. optical side lobes considered | 3 |
| Nr. iterations | 250 |

Table 4: Settings for generation of continuous transmission masks, using a non-linear least squares solver

The unit size $u$ is the minimum printable feature size. A suitable distance (in lithography context called the 'pitch') is maintained between separate primitives and features to ensure that the intended design can be produced on the wafer given the optical model properties Pang (2021). All target images are synthesized in the GDSII format that is typically used in chip design contexts Gabrielli (2020). These GDSII files are subsequently rasterized to be used in image format, and saved together with $(x, y)$ of the primitive shapes in the image plane. First, targets are converted to low resolution by applying a blurring and downsampling transformation, to make a printable image. Next, these low resolution images are zero-padded to prevent aliasing during modeling when used together with the inverse solver.

## A.4 SETTINGS FOR MASK COMPUTATION

The target images are built using the settings shown in table 3. The mask images are subsequently constructed using an inverse solver with settings as provided in table 4.

## A.5 ERROR PREDICTION MODEL ARCHITECTURE

The error prediction model uses two or three encoder blocks, based on the configuration used (difference image embedded or not). The structure of these blocks is given in the first half of Table 5; input size is (1, 256, 256). The outputs from each of the encoder blocks is a vector of size (256, 1, 1). These vectors are combined and fed through a combining convolutional block as presented in the second half of table 5. The input size is (256, 2, 1) or (256, 3, 1) depending on the configuration. As activation function after each layer LeakyReLU is used. The model is completed by two heads for regression and classification, structure of which is given in table 6. Both are fully connected neural networks, take as input a vector of length 256 and output is a single number.

## A.6 MASK PREDICTION MODEL ARCHITECTURE

The mask prediction model can be used in either an image-to-image or primitive setup. The primitive setup uses a sinusoidal encoding scheme to produce vectors for absolute and relative position. The encoding scheme has depth 200 and outputs encoded vectors of length 64. These are added to the outputs of convolutional layers. The primitive embedding scheme is given in table 9. The image embedding scheme is given in table 8. All layers in both the primitive and image embedding modules have LeakyReLU activation. Input size for the primitives is $(1, 64, 64)$ and for the images is $(1, 256, 256)$. They output a vector of length 256. For the sampling and flow setup, a mean and variation vector of size 64 are extracted by two separate linear layers. These are used to sample a vector of latent variables which is fed into the flow network. For the setup without sampling or flow, one linear layer of size 64 is used, with a LeakyReLU activation. The output of this layer is the vector of latent variables. After reparameterization, mean and variance vectors are clamped to

| layer type | nr. filters | kernel size | stride | padding |
|---|---|---|---|---|
| **Embedding modules** | | | | |
| Conv2D | 16 | [5, 5] | 1 | 2 |
| Conv2D | 32 | [5, 5] | 4 | 2 |
| Conv2D | 64 | [5, 5] | 2 | 2 |
| Conv2D | 128 | [5, 5] | 4 | 2 |
| Conv2D | 256 | [8, 8] | 1 | 0 |
| **Combining embeddings** | | | | |
| layer type | nr. filters | kernel size | stride | padding |
| Conv2D | 512 | [2, 1] or [3, 1] | 1 | 0 |
| Conv2D | 512 | [1, 1] | 1 | 0 |
| Conv2D | 256 | [1, 1] | 1 | 0 |

Table 5: Embedding module for error prediction module

| layer type | size | activation |
|---|---|---|
| **Regression** | | |
| Linear | 128 | LeakyReLU |
| Conv2D | 64 | LeakyReLU |
| Conv2D | 1 | ReLU |
| **Classification** | | |
| Linear | 128 | LeakyReLU |
| Conv2D | 64 | LeakyReLU |
| Conv2D | 1 | Sigmoid |

Table 6: Regression and classification heads for error prediction model

ranges of $[-20, 20]$ and $[-5, 5]$ respectively, for stable convergence. The final latent vectors of size 64 are fed into decoding modules that are used for mask and target prediction. Their architecture is given in table 10.

A.7   OPTIMIZATION SETTINGS ERROR PREDICTION MODEL

Optimization settings for training of the error prediction model can be found in table 11. MSE loss and BCE loss are weighted $1 : 0.1$ respectively. The ground truth for regression loss is computed as follows: we compute the maximum over all samples of taking the score of each wafer target image with respect to a zero image. Then we compute the maximum of taking MSE score of each reconstructed target image (both blurred and not blurred) with respect to a zero image. We approximate that the maximum error is the sum of these. As minimum error we take the minimum of a MSE scores over all wafer targets with respect to their reconstructed target images. Then we normalize all other scores with respect to these minimal and maximal errors.

A.8   OPTIMIZATION SETTINGS MASK PREDICTION MODEL

Optimization settings for training of the mask prediction model can be found in table 12. Loss terms are weighted as $\text{MSE}_\omega : \text{KL} : \text{MSE}_\psi = 0.5 : 10^{-10} : 0.5$. For training of the loss model, after a sensibility check that $\text{MSE}_\psi < 0.05$, the weight of $\text{MSE}_\psi$ was replaced with an exponentially decaying ($w = 0.986^{\text{epoch nr.}}$) factor. Simultaneously a predicted regression loss score was clamped between 0 and 1, and added to the total loss with weight $10^{-3} \cdot (1 - 0.986^{\text{epoch nr.}})$. Settings for flow are left at default from (Van Den Berg et al., 2018).

A.9   ABLATION STUDIES

A set of ablation studies is performed with results shown in Table 13. Models are evaluated on reconstruction towards both wafer target $\omega$ and mask $\psi$ domain. The metric used for wafer target reconstruction is the mean absolute error between ground truth $\phi$ and predicted output $\omega$. For the ILT targets (masks), the score metric is the mean absolute error between the wafer target (in low

| layer type | nr. filters | kernel size | stride | padding |
|---|---|---|---|---|
| GatedConv2D | 16 | [5, 5] | 1 | 2 |
| GatedConv2D | 32 | [5, 5] | 2 | 2 |
| GatedConv2D | 32 | [5, 5] | 2 | 2 |
| GatedConv2D | 64 | [5, 5] | 1 | 2 |
| repeat each position vector 4 times, reshape to (1, 16, 16), and concatenate with filters | | | | |
| GatedConv2D | 64 | [5, 5] | 1 | 2 |
| GatedConv2D | 64 | [5, 5] | 2 | 2 |
| GatedConv2D | 64 | [5, 5] | 1 | 2 |
| reshape each position vector to (1, 8, 8), and concatenate with filters | | | | |
| GatedConv2D | 64 | [5, 5] | 1 | 2 |
| GatedConv2D | 256 | [8, 8] | 1 | 0 |
| reshape the four position vectors to (256, 1, 1), and concatenate with filters | | | | |
| GatedConv2D | 256 | [2, 1] | 1 | 0 |

| layer type | size | activation |
|---|---|---|
| Linear | 512 | LeakyReLU |
| Linear | 256 | - |

Table 7: Primitive embedding module for mask prediction model

| layer type | nr. filters | kernel size | stride | padding |
|---|---|---|---|---|
| **Embedding modules** | | | | |
| GatedConv2D | 16 | [5, 5] | 1 | 2 |
| GatedConv2D | 16 | [5, 5] | 2 | 2 |
| GatedConv2D | 32 | [5, 5] | 2 | 2 |
| GatedConv2D | 32 | [5, 5] | 1 | 2 |
| GatedConv2D | 32 | [5, 5] | 2 | 2 |
| GatedConv2D | 64 | [5, 5] | 1 | 2 |
| GatedConv2D | 64 | [5, 5] | 2 | 2 |
| GatedConv2D | 128 | [5, 5] | 2 | 2 |
| GatedConv2D | 256 | [8, 8] | 1 | 0 |

Table 8: Image embedding module for mask prediction model

| layer type | nr. filters | kernel size | stride | padding |
|---|---|---|---|---|
| GatedConv2D | 16 | [5, 5] | 1 | 2 |
| GatedConv2D | 32 | [5, 5] | 2 | 2 |
| GatedConv2D | 32 | [5, 5] | 2 | 2 |
| GatedConv2D | 64 | [5, 5] | 1 | 2 |
| repeat each position vector 4 times, reshape to (1, 16, 16), and concatenate with filters | | | | |
| GatedConv2D | 64 | [5, 5] | 1 | 2 |
| GatedConv2D | 64 | [5, 5] | 2 | 2 |
| GatedConv2D | 64 | [5, 5] | 1 | 2 |
| reshape each position vector to (1, 8, 8), and concatenate with filters | | | | |
| GatedConv2D | 64 | [5, 5] | 1 | 2 |
| GatedConv2D | 256 | [8, 8] | 1 | 0 |
| reshape the four position vectors to (256, 1, 1), and concatenate with filters | | | | |
| GatedConv2D | 256 | [2, 1] | 1 | 0 |

| layer type | size | activation |
|---|---|---|
| Linear | 512 | LeakyReLU |
| Linear | 256 | - |

Table 9: Primitive embedding module for mask prediction model

| layer type | nr. filters | kernel size | stride | padding |
|---|---|---|---|---|
| **Prediction modules** | | | | |
| GatedConv2DT | 256 | [8, 8] | 1 | 0 |
| GatedConv2DT | 128 | [5, 5] | 2 | 2 |
| GatedConv2DT | 64 | [5, 5] | 2 | 2 |
| GatedConv2DT | 64 | [5, 5] | 1 | 2 |
| GatedConv2DT | 32 | [5, 5] | 2 | 2 |
| GatedConv2DT | 32 | [5, 5] | 1 | 2 |
| GatedConv2DT | 32 | [5, 5] | 2 | 2 |
| GatedConv2DT | 16 | [5, 5] | 2 | 2 |
| GatedConv2DT | 16 | [5, 5] | 1 | 2 |
| Conv2D | 1 | [1, 1] | 1 | 0 |

Table 10: Mask and wafer target prediction modules for mask prediction model

| | |
|---|---|
| epochs (two-channel input) | 300 |
| epochs (three-channel input) | 500 |
| batch size | 32 |
| learning rate | 5e-5 |
| gradient clipping | 0.5 |
| image size | (256, 256) |
| threshold for classification | 0.5 |

Table 11: Optimization settings for error prediction model

| Configuration | Proposed | No prim. | No rel. pos. | No sampl./flow | No classifier | Benchmark |
|---|---|---|---|---|---|---|
| epochs | 200 | 200 | 200 | 200 | 1000 | 1000 |
| batch size | 32 | | | | | |
| learning rate | 1e-5 | | | | | |
| grad. clipping | 0.6 | 0.7 | 0.6 | 0.6 | 0.6 | 0.6 |
| mask im. size | (256, 256) | | | | | |
| prim. im. size | (64, 64) | - | (64, 64) | (64, 64) | (64, 64) | - |
| input im. size | - | (256, 256) | - | - | - | (256, 256) |
| flow type | orthogonal | orthogonal | orthogonal | - | orthogonal | - |

Table 12: Optimization settings for mask prediction model

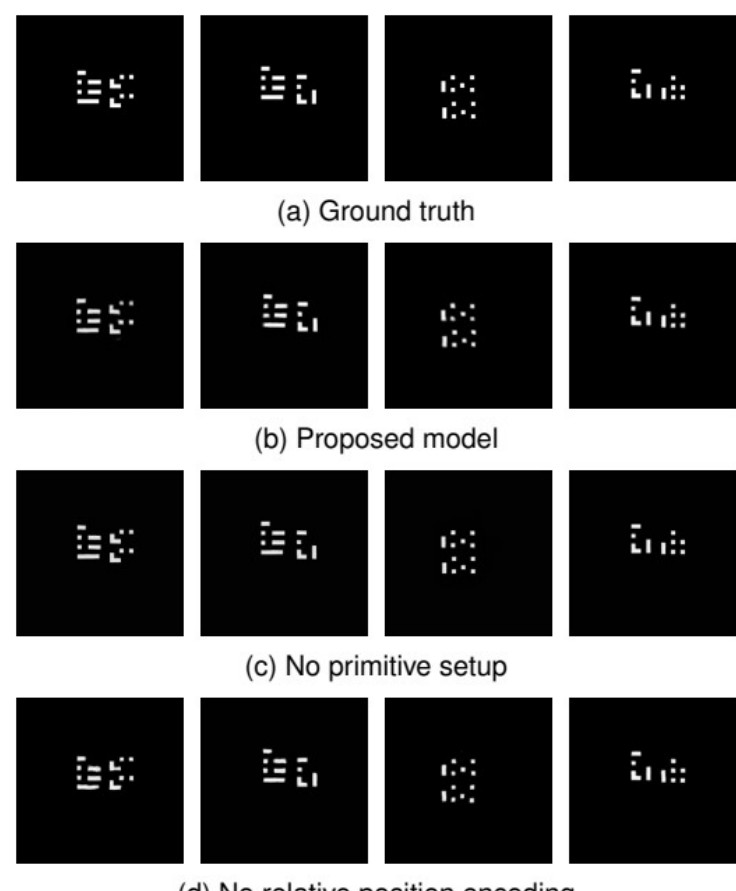

Figure 4: Wafer target reconstruction of four different samples, for different model configurations.

| Model configuration | Prediction error | | # params |
| --- | --- | --- | --- |
| | MAE$(\phi, \omega)$ | MAE$(\phi_{lr}, F(\psi))$ | |
| Full model | $0.0075 \pm 0.065$ | $0.0026 \pm 0.026$ | 13.9M |
| A: no primitives | $0.0021 \pm 0.026$ | $0.00083 \pm 0.0089$ | 14.9M |
| B: no relative pos. | $0.0061 \pm 0.059$ | $0.0023 \pm 0.024$ | 13.9M |
| C: no sampling/flow | $0.0098 \pm 0.077$ | $0.0031 \pm 0.031$ | 13.2M |
| D: no classifier loss | $0.0249 \pm 0.13$ | $0.0100 \pm 0.071$ | 13.9M |

Table 13: Results of ablation studies

resolution) $\phi_{lr}$ and the result of passing the predicted mask through the optics and resist model $F(\psi)$. Performance of the full model (the proposed method) is shown for comparison. Four extra model configurations are tested, leaving out one of our proposed methods at a time. Setup A omits the primitive encoding part, using image-to-image mapping instead. In B we use the primitive setup, but leave out the relative position encoding. C goes back to a deterministic modeling setup, with no sampling or flow in the latent space. Finally, setup D uses a naive MSE loss function on the mask prediction instead of the trained loss scoring model. Note that combining all these ablations results in the benchmark model setup. Some examples for wafer target reconstruction by different model configurations are shown in Fig. 4.

Looking at the comparison between benchmark and proposed model, we see that the benchmark performance is better. This is expected, as its task is simpler: we omit the complex primitive setup and although we will not have a spread in the predicted mask domain, the model can learn to predict a general mask shape well. In the ablation studies we see the same effect: the model performs

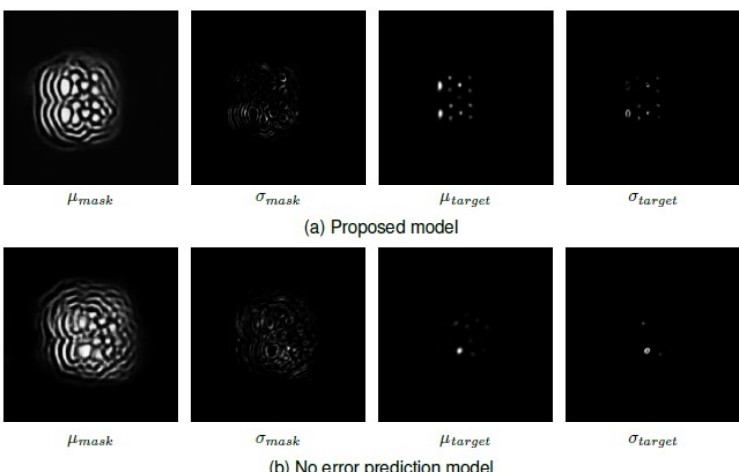

$\mu_{mask}$        $\sigma_{mask}$        $\mu_{target}$        $\sigma_{target}$

(a) Proposed model

$\mu_{mask}$        $\sigma_{mask}$        $\mu_{target}$        $\sigma_{target}$

(b) No error prediction model

Figure 5: Visualization of spread between different predictions. Columns 1 and 2 show the mean and variance of predicted masks respectively. Columns 3 and 4 show the mean and variance of predicted targets respectively.

|  | **Prediction error** | |
| --- | --- | --- |
| **Model config.** | Spread masks | Spread targets |
| A: proposed | $8.85 \cdot 10^{-4}$ | $5.28 \cdot 10^{-5}$ |
| B: benchmark | 0 | 0 |
| C: no loss model | $4.07 \cdot 10^{-3}$ | $1.89 \cdot 10^{-4}$ |

Table 14: Results of sampling multiple masks

slightly better about 3x better when omitting the primitive setup. Furthermore, the ablation studies show that adding relative position is not helpful in this setup. However, earlier results on toy datasets showed that adding relative position improved the model's reconstruction ability. There might be other methods to add context to the model, that work better in the proposed setup. Finally we see that adding the error prediction model to the setup improves performance, which is desired. However, we did not fully investigate using the exponentially decaying MSE loss without adding an error prediction model, which could also be of influence.

### A.10 MULTIPLE MASK PREDICTION

One goal of this study was to enable the one-to-many mapping in mask prediction. We test this by sampling multiple times from an encoded set of primitives, and evaluating the spread of predicted mask images. This spread $s$, is measured using mean absolute differences between predicted masks. A high spread within the set of predicted masks is only useful if the masks actually map back to one wafer target. Therefore we also evaluate the spread of the images resulting from putting the predicted masks through the optics and resist model. Results are averaged over five data samples, with 20 masks sampled from each, and shown in Table 14 for different model configurations. Fig. 5 shows a visual example of the differences in masks and wafer targets.

Comparing the spread in predicted masks between the setups with and without error prediction model, we actually see that using no error prediction model we have a higher spread in the predicted output masks. However, the prediction errors for this setup are more than twice as high, and also the spread in wafer target domain is higher which is not desired. Having multiple faulty predictions can also lead to a higher spread in the prediction domain; but this is not the desired effect. Visually looking at the results in Fig. 5 we see that differences between predictions are mostly around the edges of target and main features, which is as expected. The main features are represented quite well across all images.

