# OpenReview forum: "Primitive embeddings for generative modeling in inverse lithography"
_ICLR.cc/2026/Conference — ICLR 2026 Conference Withdrawn Submission_

### Official Review · Reviewer_cyaL · 2025-10-28

**Soundness:** 1
**Presentation:** 1
**Contribution:** 2
**Rating:** 2
**Confidence:** 5

**Summary:**

This paper deals with the mask optimization problem in in context of inverse lithography. Unlike prior arts of a normal non-convex optimization, this work formulates the mask optimization as max a posteriori. Interesting problem setting, but I doubt the practicality due to a lack of supporting experiments.

**Strengths:**

Observe the ill-posed setting in mask optimization and formulate the problem from a statistical point of view.

**Weaknesses:**

There are many weaknesses that make the paper less ready for publication. Please find my detailed comments below:
1. My biggest concern is the experiments, in the semiconductor manufacturing industry, there are already well-defined metrics to evaluate mask optimization: EPE, process windows, Depth of Focus, LWR, LER, MRC, etc. None of these are displayed in the paper to support the method.
2. The writing needs significant improve. I have no clue what the model is doing in Fig.1 and Fig. 2. What are the inputs what are the outputs. If I give you a chip design after PnR, how do I use your model to get the optimized mask?
3. There are many inaccurate descriptions in the paper. I recommend that the authors have a more comprehensive literature survey, e.g.:
a. "In addition, changing a single feature still requires remapping the entire patch." Not necessarily true, the scope is defined by the abmit or optical diameter. the changing feature does not propagate that far.
b. Please include few ill-posed examples, for mask optimization, since we never reach global optimal due to the low-pass filter of lithography system, we will simply pick the best based on the metrics.
c."The space of possible masks Sϕ for full target images is vast and searching it naively with an optimization procedure will take a long time."Not ture, mask is limited by MRC's, so it has quite structural similarity to the target design pattern except SRAFs.
4. There are lots of ILT/OPC papers cited, but there should be comprehensive experiment comparisons with SOTA to support the methodology.

**Questions:**

refer to the weakness.

---

### Official Review · Reviewer_1CFY · 2025-10-30

**Soundness:** 2
**Presentation:** 2
**Contribution:** 2
**Rating:** 2
**Confidence:** 4

**Summary:**

This manuscript proposes a generative modeling method based on primitive embeddings to address the mask design problem in Inverse Lithography Technology (ILT). The contributions of the paper include: (1) introducing a primitive modeling approach that represents chip designs as combinations of basic geometric shapes to enable more flexible and modular design; (2) proposing the use of a variational autoencoder combined with normalizing flows to model the one-to-many mapping relationship from wafer targets to masks; (3) designing an error prediction model to evaluate the joint plausibility between the mask and the wafer target. The method is evaluated on a synthetic dataset and compared against an image-to-image baseline. Although primitive modeling offers potential advantages in application scenarios, the experimental results indicate that the performance of the proposed method in its current setup still underperforms the baseline approach.

**Strengths:**

(1) The use of primitive modeling is a novel approach; this modular design workflow is more flexible and has the potential to resolve inconsistencies caused by patching.

(2) The authors appropriately formulate the problem as a maximum a posteriori estimation in a probabilistic generative model, and the use of a VAE architecture combined with normalizing flows for modeling is technically sound.

(3) Designing an independent error prediction model to evaluate the joint plausibility between the mask and the wafer target is an innovative idea.

**Weaknesses:**

(1) In Table 2, the proposed method significantly underperforms the baseline model across all evaluation metrics. The performance gap in MAE between the proposed model and the baseline is too substantial to be acceptable for a deep learning conference paper. Although the authors claim the results are "promising from an application point of view" (in the Abstract), this does not compensate for the large performance discrepancy.

(2) While primitive embedding is the core contribution of this paper, the experimental results contradict this claim. The results fail to support this viewpoint, as Table 13 shows that performance clearly improves after removing primitives. Does this indicate that primitive embedding, as an input representation, actually harms model performance?

(3) The authors only use four types of primitives, which may be insufficient compared to real-world complex IC manufacturing scenarios. The authors should discuss whether more primitive types should be incorporated.

(4) All experiments are conducted on synthetic data, with no validation on any real industrial datasets, which is a serious limitation. The authors also mention that the current dataset is relatively simple. Can the current "success" only be achieved on the authors' simple synthetic dataset? We doubt whether the model possesses sufficient generalization capability.

(5) The authors only compare their method with a simple baseline model, without comparisons against other proposed methods. This makes it difficult for readers to evaluate the effectiveness of the proposed approach.

(6)The authors conclude that their method saves computational time and resources, but they do not report training time, inference time, or computational resource requirements.

**Questions:**

In Table 13, model performance improves after removing positional encoding. The authors do not explain this phenomenon. Is positional encoding truly beneficial?

**Details Of Ethics Concerns:**

No.

---

### Official Review · Reviewer_WvVb · 2025-10-31

**Soundness:** 2
**Presentation:** 1
**Contribution:** 1
**Rating:** 2
**Confidence:** 3

**Summary:**

This paper introduces a primitive-based modeling setup with a variational autoencoder (VAE) architecture and introduce a separable feature-encoding scheme for flexible target generation and straightforward domain validation.
To address the one-to-many mapping from targets to masks, it employs generative modeling and introduce a novel scoring strategy to effectively train under this probabilistic formulation.

**Strengths:**

This paper tries to apply generative modeling to the inverse lithography process in chip manufacturing, which is a reasonable idea.
They do some modeling of the problem, which is also reasonable.

**Weaknesses:**

1. Contributions are not explicitly listed.
1. Figures are lacking and poorly done. Instead, space that could be used to give insight via illustration is spent on formulas that are thoroughly covered in other works.
1. This paper only compares against one benchmark, it does not achieve benchmark-level accuracy [line 029]. Reference to the  benchmark used [line 374] is without reference to the appendix, leaving it mysterious to the reader.
1. This paper provides synthetic data generation details in the appendix but lacks a concise explanation in the main text [line 316].
1. In Figure 1, the components of the primitive embedding module are not described, and Section 5.1 (Primitive Embedding) also lacks detailed explanation [line 216--262].
1.  The experimental results are limited in scope and fail to demonstrate competitive performance [line 412].

This seems to be incomplete - there is an idea, but it has not been shown to work.  It has not been thoroughly compared to other approaches, and the simpler benchmark outperforming it undermines their contribution.

**Questions:**

1. Should the error prediction model compute difference: |𝟇 - 𝟂| instead of |𝟂 - 𝟁| [line 306]?
2. Why is the size of $Z_{0, i}$  64 [line 284]? How was 64 chosen?

---

### Note · Authors · 2025-12-01

**Comment:**

Having read the reviews, we have not enough time nor space to sufficiently improve this paper. I would like to thank the reviewers for their valuable feedback.

**Withdrawal Confirmation:**

I have read and agree with the venue's withdrawal policy on behalf of myself and my co-authors.